# Pelvic Exenteration for Recurrent Vulvar Cancer: A Retrospective Study

**DOI:** 10.3390/cancers16020276

**Published:** 2024-01-08

**Authors:** Sabrina Classen-von Spee, Saher Baransi, Nando Fix, Friederike Rawert, Verónica Luengas-Würzinger, Ruth Lippert, Michelle Bonin-Hennig, Peter Mallmann, Björn Lampe

**Affiliations:** 1Department of Gynecology and Obstetrics, Florence-Nightingale-Hospital, Kreuzbergstraße 79, 40489 Düsseldorf, Germany; baransi@kaiserswerther-diakonie.de (S.B.); fixn@kaiserswerther-diakonie.de (N.F.); rawert@kaiserswerther-diakonie.de (F.R.); luengas-wuerzinger@kaiserswerther-diakonie.de (V.L.-W.); bonin@kaiserswerther-diakonie.de (M.B.-H.); 2Department of Pathology, Evangelisches Krankenhaus Oberhausen, Virchowstraße 20, 46047 Oberhausen, Germany; 3Department of Obstetrics and Gynecology, University of Cologne, Kerpener Str. 62, 50937 Cologne, Germany; peter.mallmann@uk-koeln.de

**Keywords:** recurrent vulvar cancer, pelvic exenteration, overall survival, complications, p53, p16/HPV

## Abstract

**Simple Summary:**

Almost half of patients with vulvar cancer develop a recurrence after primary surgical treatment. There is no clear recommendation for treatment, especially in advanced cases. Data for chemotherapy/radio(chemo)therapy are rare, with moderate to poor response rates and overall survival. Additionally, due to radiation within the primary treatment, radio(chemo)therapy is usually not a possible treatment method. Hence, pelvic exenteration (PE) might be the only choice for curation, with good overall survival and moderate morbidity. PE showed improved quality of life, even in palliative situations, especially in cases of cloacas or urogenital/intestinal fistulas. There are only a few studies analysing PE in vulvar cancer exclusively, and none of them are dealing with recurrent cases separately. Due to this, we analysed 17 cases of recurrent vulvar cancer that underwent PE in our department. The aim was to identify relevant histopathological and clinical factors for overall survival. To further analyse PE for recurrent vulvar cancer, a comparison with the existing literature on PE and radio(chemo)therapy in recurrent vulvar cancer has been performed.

**Abstract:**

Pelvic exenteration (PE) is one of the most radical surgical approaches. In earlier times, PE was associated with high morbidity and mortality. Nowadays, due to improved selection of suitable patients, perioperative settings, and postoperative care, patients’ outcomes have been optimized. To investigate patients’ outcomes and identify possible influencing clinical and histopathological factors, we analysed 17 patients with recurrent vulvar cancer who underwent PE in our department between 2007 and 2022. The median age was 64.9 years, with a difference of 40 years between the youngest and the oldest patient (41 vs. 81 years). The mean overall survival time was 55.7 months; the longest survival time reached up to 164 months. The achievement of complete cytoreduction (*p* = 0.02), the indication for surgery (curative vs. palliative), and the presence of distant metastases (both *p* = 0.01) showed a significant impact on overall survival. The presence of lymphatic metastases (*p* = 0.11) seems to have an influence on overall survival (OS) time. Major complications appeared in 35% of the patients. Our results support the existing data for PE in cases of recurrent vulvar cancer; for a group of selected patients, PE is a treatment option with good overall survival times and acceptable morbidity.

## 1. Introduction

In the last few years, there has been a significant overall increase in the incidence of vulvar cancer in high-income countries [1]. Vulvar cancer arises from two premalignant lesions: dVIN (differentiated vulvar intraepithelial neoplasia) resulting from chronic inflammatory conditions such as Lichen sclerosus or Lichen Planus and uVIN (usual-type vulvar intraepithelial neoplasia)/VHSIL (vulvar high-grade squamous intraepithelial lesions), resulting from HPV-(Human papillomaviruses) infections [2,3,4,5]. The two entities show different pathophysiologies with diverse treatments and prognosis, resulting in a different risk of progressing to an invasive carcinoma. The 10-year risk for cancer development is about 60% for dVIN vs. 10% for uVIN [6]. For dVIN, a complete surgical excision is recommended, and the underlying Lichen sclerosus/Lichen Planus should be treated with topical high-potency corticosteroids [7]. In the case of VHSIL, excisional, ablative, or medical (e.g., imiquimod) treatments can be considered [7].

Surgery represents the basic therapy for primary vulvar cancer. Surgical treatment, usually consisting of wide excision or radical vulvectomy, with or without subsequent radiotherapy, is associated with an excellent prognosis in the early stages of vulvar cancer. However, because of delayed symptoms and consecutive late diagnoses, approximately 40% of the cases show an advanced stage (FIGO III or IV) [8,9]. These advanced cases require a radical approach to treatment. The involvement of lymphatic nodes is a strong prognostic factor for the outcome of vulvar cancer [10,11,12,13]. The probability of lymph node involvement and the number of lymph nodes affected remained constant over the last 40 years, indicating impaired detection of vulvar cancer in earlier stages [14].

About 12–37% of all cases develop recurrent disease within the first two years [15], and 40–50% of the patients develop recurrence after surgical treatment regardless of the time [16].

There is no clear recommendation for the treatment of recurrent vulvar cancer, especially in advanced cases [17]. Data for chemotherapy alone are rare, with a moderate to poor response rate and overall survival [18]. Radio(chemo)therapy is usually not an option due to the already extensive radiotherapy during primary treatment, the risk of fistula formation in cases of advanced carcinomas, or the extent of the tumor mass. Therefore, surgery is often the only remaining choice for adequate treatment.

Progressed recurrent as well as primary advanced carcinomas of the vulva are often close to or are already infiltrating adjacent organs (vagina, urethra, anus, rectum). This can result in uro- and intestinal-genital fistulas (especially in cases with previous radiation) or cloacas, as shown in Figure 1, reducing the quality of life and leaving physicians with limited treatment options. In these cases, organ-saving surgery might not be possible, even after neoadjuvant radio(chemo)therapy. Hence, for primary advanced and progressed recurrent tumors or in cases not feasible for radio(chemo)therapy, pelvic exenteration (PE) could be performed with a curative or palliative intention. Only a few studies have been published addressing PE in the treatment of vulvar cancer exclusively [19,20,21,22]. In all studies, primary and recurrent tumors were analyzed together, whereas this study analyzes PE in recurrent vulva cancer exclusively. Beside Forner et al. [22], our data represents the largest cohort of patients with recurrent vulvar cancer undergoing PE. The aim of this study was to identify relevant histopathological and clinical factors for overall survival and to analyze perioperative complications.

## 2. Materials and Methods

### 2.1. Study Design and Population

The aim of this study was to perform a retrospective data analysis of patients with recurrent vulvar cancer who underwent PE at the Florence-Nightingale-Hospital (FNK) in Duesseldorf, Germany, between 2007 and 2022. Only recurrent squamous cell cancers (SCC) were included. To keep the cohort homogenous, other histological subtypes were excluded. Recurrent cancer was defined as a locoregional or distant relapse. PE was defined as the resection of the vagina, uterus, ovaries, and fallopian tubes; for anterior PE, the bladder; for posterior PE, the rectum; or a combination of both (total exenteration). Patients’ data were collected from an internal database with access to clinical, surgical, and histological reports. The primary endpoint was overall survival (OS). To report OS, recent patient data were obtained.

### 2.2. Statistical Analysis

Patients’ data were evaluated by calculating the mean with the interquartile range. The primary endpoint was OS, which was defined as the time (in months) between surgery and the date of death (of any cause) or the date of last contact. Survival time (in months) after surgery was clarified as a dependent variable. To identify prognostic factors, Kaplan–Meier analyses as well as Cox regression analyses were performed to estimate survival risks and hazard ratios. For statistical analyses, IBM SPSS Statistics 29 was used (IBM Corp., Released 2021, IBM SPSS Statistics for Windows, Version 28.0, Armonk, NY, USA: IBM Corp.). The following independent variables were analyzed: indication for surgery (palliative, curative), achievement of complete cytoreduction (R1 vs. R0), tumor grading (G1/G2/G3), stadium (according to the recommendations of the World Health Organization (WHO) and the International Federation of Gynecology and Obstetrics (FIGO), involvement of lymphatic nodes/lymphatic vessels/blood vessels, p53- and p16-mutation status, and postoperative complications (Clavien–Dindo classification [23]). In cases where R0-resections were unachievable and in cases with distant metastases (pM1/cM1), the indication for surgery was defined as palliative. If R0-resection seemed achievable, the indication was defined as curative. 

### 2.3. Assessment of Perioperative Morbidity

Perioperative morbidity was defined following the Clavien–Dindo classification [23]; the level of the complication depends on the therapy required. Major morbidity was defined as Clavien–Dindo ≥ 3.

### 2.4. Immunohistochemistry

For the immunohistochemistry (IHC) analysis, blank 4 µm sections were cut from formalin-fixed and paraffin-embedded tissue and stained with p16-antibody (Anti-p16 (E6H4), Ventana; Platform: Ventana BenchMark, Tucson, AZ, USA) and p53-antibody (Anti-p53, Clone DO-7, Ventana; Platform: Ventana BenchMark, Tucson, AZ, USA).

p53 showed a wild-type expression when cells had a mosaic-like nuclear expression (Figure 2C). An over-expression with accumulation of the protein and consecutive strong staining of nearly all tumor cell nuclei (gain of function, Figure 2B) or complete loss of the staining (loss of function, Figure 2A) were recorded as mutations.

The expression of p16 was recorded as wild-type/normal when tumor cells showed no or only a weak expression (Figure 2E). Cells with an abnormally strong or block-type expression of p16 were considered abnormal and therefore positive for HPV-infection (Figure 2D).

### 2.5. Literature Review

To compare our results with the published literature, the Pubmed/MEDLINE and the Cochrane Library databases were searched for studies analyzing PE for the treatment of vulvar cancer. To filter the results, a useful combination of MESH-terms was used, and the analysis of the literature was performed with reference to the Preferred Reporting Items for Systematic Reviews and Meta-Analyses (PRISMA). The last date of retrieval was 19th August 2023. Only clinical trials, meta-analyses, and systematic reviews in English/German were included; further inclusion criteria were analyses of squamous cell tumors of the vulva exclusively, open exenteration for recurrent, progressive, or advanced disease, and evaluation of the data utilizing Kaplan–Meier analysis.

Case reports and studies without a detailed description of the collective or studies describing collectives of mixed gynecological cancers were excluded, as well as studies considering laparoscopic or robotic PE.

## 3. Results

### 3.1. Patient Data Analysis

Seventeen patients underwent PE in the FNK during 2007 and 2022 due to recurrent squamous cell tumors of the vulva. All of them showed local recurrence; two patients additionally had distant metastases. Patient and tumor characteristics are summarized in Table 1.

The median age was 64.9 years, with a difference of 40 years between the youngest and the oldest patient (41 vs. 81 years). Three patients (17%) underwent total, four patients (24%) posterior, and ten patients (59%) anterior PE. Seven patients (41%) received reconstructive surgery of the vulva. Eleven patients (64%) received a PE during treatment for the first recurrence. The second, third, and fourth recurrences were treated in two cases (12%) each. Five patients (28%) presented with rT2-, ten patients (59%) with rT3-, and two patients (12%) with rT4-tumors. Lymph nodes were simultaneously removed in nine cases (53%); positive lymph nodes appeared in two cases (12%); in seven cases (41%), no lymphatic metastases were found. In one case (6%), a metastasis of the vulvar skin was diagnosed (pM1), and in another case (6%), a metastasis of the lung was found radiologically (cM1). These two (12%) patients had a palliative indication for PE; the other 15 (88%) cases had a curative indication. Invasion of lymphatic vessels was detected in three patients (17%), and invasion of blood vessels in one case (6%). In six cases (35%), complete cytoreduction could not be reached during PE. In two (12%) cases, a poorly differentiated carcinoma (G3) was found; the other 15 (88%) tumors showed moderate differentiation (G2). There was no case with good differentiation (G1). Two patients (12%) had received neoadjuvant radiochemotherapy; three patients (17%) received adjuvant therapy after PE. The expression of p16 and p53 was analyzed by IHC (Table 1). Eleven patients (64%) showed an aberrant expression of p53 and a normal expression of p16. p16 showed an overexpression in three cases (17%), but no aberrant expression could be found in two cases (12%). In one case, the mutation-status is unknown.

### 3.2. Data Analysis

Figure 3 shows the overall survival in months for all 17 patients with recurrent squamous vulvar cancer. The mean survival time was 55.7 months, with a minimal survival of one and a maximal survival of 164 months after PE. Ten patients died; two patients were still alive at the time of the last follow-up (July 2023). There was a loss of follow-up for five patients; the last contact was between two and ten months after surgery.

Survival was influenced by the achievement of complete cytoreduction, by the presence of distant metastases (which led to a palliative indication for surgery), and by the presence of lymphatic metastases.

The mean OS after achieving R0-resection was 69.5 months, ranging from three to 164 months. In cases of R1-resection, the mean survival was 6.6 months, with a range of one to nine months (Figure 4; Log rank: χ^2^(1) = 5.44, *p* = 0.02).

The mean survival in cases of distant metastases was four months (Figure 5). In the case of cM0, the mean survival time was 59.4 months (Log rank: χ^2^(1) = 6.50, *p* = 0.01). The two patients with distant metastases were the only cases of palliative surgery.

Seven patients underwent the removal of lymphatic nodes: in one case, the inguinal nodes, and in six cases, the pelvic lymphatic nodes. In one case, the paraaortic lymphatic nodes were removed additionally. Only two patients showed lymphatic metastases; in one case, the inguinal and in the other case, the pelvic lymphatic nodes were affected. The analysis of the presence of lymphatic metastases revealed an effect on the OS with a mean survival of four months for patients with metastases and a mean survival of 59.0 months for patients without metastases (Figure 6; Log rank: χ^2^(1) = 2.51, *p* = 0.11).

No statistically significant effect on survival time was seen based on age, tumor stadium, tumor grade, lymphatic and blood vessel invasion, postoperative complications, or mutations of p16 or p53. 

The results of the univariate analysis are summarized in Table 2. Due to the small number of cases, a multivariate analysis was not performed.

### 3.3. Complications

Postoperative complications classified by the Clavien–Dindo classification [23] appeared in every patient, with minor complications in eleven patients (65%) and major complications (Clavien–Dindo ≥ 3) in six patients (35%). Seven patients (41%) showed grade 1 complications (e.g., need for electrolytes, analgesics, and physiotherapy); four patients (24%) had grade 2 complications (e.g., need for blood transfusion). In five cases (29%), a surgical intervention with anesthesia was necessary (Clavien–Dindo 3b), and one patient (6%) required dialysis (Clavien–Dindo 4a) during the postoperative period. Table 3 gives an overview of complications in all 17 patients. 

Patients with complications classified as ≤2 showed a mean overall survival time of 63.3 months. In contrast to complications classified >2, with a mean OS of 19.6 months, the difference in OS was not statistically significant (Log rank: χ^2^(1) = 0.51, *p* = 0.48).

### 3.4. Literature Review

Using the Mesh term “pelvic exenteration vulvar cancer”, 182 publications were found (1956–2023); 162 of those were written in English/German, including 24 case reports. After filtering and adaptation of the inclusion/exclusion criteria, four publications remained (Table 4). Controlled randomized studies or studies including a control group were missing.

## 4. Discussion

Treatment options for recurrent vulvar cancer are limited as there are no evidence-based standards for systemic treatment, especially in advanced cases [6]. Treatment options strongly depend on tumor size, its localization, and the previous therapy. Most of the recurrent tumors appear in the vulvar region; advanced recurrent disease might affect the vagina, urethra, bladder, and rectum, as well as neurologic or osseous structures within the pelvis. Recurrences in the groin or in distant organs rarely appear later than two years after the primary diagnosis [17].

Following ESGO 2023 [17], for local, limited recurrent tumors, radical excision is suggested. In locally advanced stages, the ESGO 2023 recommends definitive radiochemotherapy in radiotherapy-naive patients. PE is recommended in selected cases, especially when radiotherapy is not an option. Following the German guidelines for vulvar cancer, which are currently under revision, in cases of locoregional recurrence, radical excision is recommended [24]. In cases of unresectable tumors, radiochemotherapy should be preferred, and if this is not possible, palliative therapy is recommended, which is not defined in detail. The National Cancer Institute recommends a wide local excision with or without radiation in cases of local recurrence [25]. In advanced cases, a radical vulvectomy, or PE, is approved. Furthermore, synchronous radiation and chemotherapy, with or without surgery, are recommended.

As most of the patients are not radiotherapy naive or have a huge tumor volume unattainable for radiation, PE is an important treatment option, as it is often the only chance for curation. Nevertheless, not much data, considering PE exclusively in recurrent vulva cancer, have been published.

Initially, performing a meta-analysis of PE in cases of vulvar cancer was planned. Data analysis showed a lack of randomized controlled trials, studies containing a control group, or studies of (recurrent) vulva cancer exclusively. Furthermore, most of the studies did not describe the patient groups in detail; hence, no meta-analysis could be performed. Therefore, we exclusively analyzed the data of 17 women with recurrent vulvar cancer who underwent PE in our department, exploring relevant histopathological and clinical factors for overall survival.

The mean OS was 55.67 months (4.6 years), with a major complication rate (Clavien–Dindo Classification ≥ 3b [23]) of 35%. The resection status was a significant factor influencing overall survival (*p* = 0.02), as described in other studies, too. In our cohort, R0-resection led to a mean OS of 69.5 months, while in cases of R1-resection, the mean OS was 6.6 months. Distant metastases (*p* = 0.01) and indication for PE (*p* = 0.01) showed statistically significant influence on the survival time. Even if just two patients had distant metastases, resulting in a palliative indication for surgery, these factors seem to have a high prognostic value for reduced survival. The presence of distant metastases has not been described as a negative factor for patients’ outcomes in cases of PE in vulvar cancer before. This might be due to the exclusion of patients with distant metastases in other studies dealing with PE. Overall, it seems logical that distant metastases, as well as a palliative indication for surgery, are associated with a shorter overall survival. Nevertheless, palliative PE can improve patients’ quality of life and should be considered in selected cases.

The presence of lymphatic metastases showed an impact on patients’ survival (*p* = 0.11), as lymphatic metastases are already described to be associated with worse OS in patients with PE in cases of vulvar cancer [21,22]. The impact was not significant, probably due to the small number of patients with lymphatic metastases (*n* = 2).

All 17 patients showed postoperative complications; eleven cases (65%) showed minor complications, and six patients (35%) showed major complications according to the Clavien–Dindo Classification [23]. Since grade 1 complications include the necessity of analgetics, electrolytes, and physiotherapy and grade 2 complications involve the need for blood transfusion, it is not remarkable that every patient had a postoperative complication according to the Clavien–Dindo Classification subsequent to such an extensive surgical intervention as PE.

As vulvar SCC can be categorized into three histological subgroups (HPV-positive, HPV-negative/p53 mutant, and HPV-negative/p53 wildtype), we analyzed the expression of p16 and p53 by IHC. HPV-association is characterized by p16 overexpression and occurs in about 40% of vulvar cancers. This type of cancer is mostly detected in younger patients (age 30–50 years), and the 5-year overall survival rate is about 83% [26,27,28,29]. It is known that HPV-independent carcinomas usually occur in post-menopausal patients. These tumors often harbor an aberrant p53 expression, leading to a high rate of local recurrence and a worse 5-year OS of approximately 48% [2,28,30]. 

In our cohort, 11 (64%) patients showed the HPV-negative/p53 mutant subtype, and three cases were HPV-positive (17%). No aberrant expression (HPV-negative/p53 wildtype) was found in another two cases (12%). This distribution was very similar to the result of Kortekaas et al. [26], where 66% of the cases were HPV-negative/p53 mutant, 16% were HPV-negative/p53 wildtype, and 18% were HPV-positive. The results of Kortekaas [26] showed a worse outcome for HPV-negative/p53 mutant tumors, while HPV-positive tumors had a favorable outcome (5 years OS: 83 vs. 48%, 14). In our study, mutations of p16 or p53 did not show an effect on survival. This may be due to the small number of patients.

By data bank research, four studies dealing with PE in cases of SCC of the vulvar exclusively were found [19,20,21,22] (Table 4; for inclusion and exclusion criteria, see above). All four studies included primary, advanced, and recurrent cases. Overall, a 5-year OS between 38 and 69% was described. In a single-center retrospective analysis, Valstad et al. (2023) [19] described a group of 30 patients with primary (53%) and recurrent (47%) squamous cell vulvar cancer who underwent PE in Norway. The complication rate was described as 90-day morbidity for grade 3 complications (63%); mainly wound infections were reported. Approximately 7% of the patients had no complications. The 5-year OS was 50%, and the median overall survival was 5.05 years. No significant differences between cases with primary and recurrent disease were found. Furthermore, no significant correlation could be identified in the uni- and multivariate analysis. Abdulrahman et al. 2022 [20] analyzed 19 patients with primary locally advanced (*n* = 14) and recurrent (*n* = 5) SCC of the vulva, treated with PE. Macroscopic tumor clearance was achieved in all patients; microscopic tumor clearance could not be achieved in one case with recurrent vulvar cancer. The described 30-day major (Clavien–Dindo Classification ≥3) morbidity rate was 42%. In the entire group, the 5-year overall survival rate was 66.7%, with a mean OS of 144.8 months (2–206 months). For the group with primary disease, the 5-year OS was 69.3% vs. 60% for the group with recurrent cancer. The mean overall survival showed a marked difference for both groups (primary cancer, 152.2 months vs. recurrent cancer, 45.8 months). In the group of primary disease, the lymphovascular invasion could be described as a significant prognostic factor for OS (36.5 vs. 182.1 months); this result was not found in the group of recurrent diseases; the reason for this could be the small number of patients in this group (*n* = 5). The occurrence of perineural invasion was associated with a trend toward a poor prognosis.

Hopkins et al. 1992 [21] analyzed 19 patients with primary, advanced (*n* = 11), and recurrent (*n* = 8) SCC. 52% of the patients showed major complications. The 5-year survival was 60%, which was significantly influenced by the status of the lymphatic nodes (*p* = 0.02). The survival rate was 64% in cases of primary cancer and 38% in cases of recurrence.

Forner et al. (2011) [22] described 27 cases of vulvar cancer with primary (*n* = 9) and recurrent (*n* = 18) disease. All patients showed no macroscopic residual tumor after surgery; 74% of the cases showed no microscopic residues (R0). Approximately 67% of the patients had postoperative complications, of which 44% were minor complications and 22% required surgical intervention. The median survival time was 37 months, the 5-year survival was 62%, and the overall survival was 59%. Involvement of the lymphatic nodes showed a significant impact on survival (OS 40 vs. 76%, 5-year OS 36 vs. 83%) as well as the resection status (5-year survival 74 (R0) vs. 21 (R1). There was no significant difference in the outcome of cases with primary and recurrent disease (OS 56 vs. 61%, 5-year OS 62 vs. 59%).

Our study and the four studies mentioned showed good results for PE in cases of vulvar cancer with acceptable rates of major complications. In conclusion, PE is a legitimate treatment for women with recurrent—and primary; locally advanced—vulvar cancer. The results show that tumor-free margins are a significant marker for better OS; hence, total tumor resection should be aspired to in cases of PE, if possible. Forner et al. [22] and Valsted et al. [19] showed no difference in mean OS between recurrent and primary disease. Abdulrahman et al. and Hopkins et al. showed reduced OS in recurrent cases; this could possibly be explained by different patients’ and tumor characteristics.

Our study is only partially comparable to the studies mentioned, as our collective consists of recurrent cases only. The resection status seems to be a reliable factor for the mean OS. In the mentioned studies, the presence of lymphatic metastases and the invasion of lymphatic vessels also showed a significant influence on the OS. In further literature, lymphovascular space invasion and perineural invasion have been described as predictors of poor prognosis [31,32,33,34].

Beside surgical treatment, systemic therapy (e.g., chemotherapy and/or radiotherapy) is an option for recurrent vulvar cancer. As mentioned before, there are no distinct recommendations regarding the therapy of recurrent vulvar cancer, but as 40–50% of the patients with primary vulvar cancer develop a recurrence after initial surgical treatment, therapeutic standards for recurrent disease are needed [16].

Comprehensive studies or clinical trials comparing the results of surgical treatment vs. radio(chemo)therapy are missing. Additionally, most of the existing studies are focused on systemic or radio(chemo)therapy for primary advanced cancer, and this data could lead to treatment suggestions for recurrent cancers. Overall, vulvar cancer is rarely studied exclusively. Due to the small case numbers, most studies analyzed data from vulvar cancer together with vaginal or cervical cancer. Hence, some recommendations for treatment of vulvar cancer are based on treatment for cervical cancer.

For chemoradiation, the reported complete response rates are low (50% or less), the 5-year OS is limited to 30–60% [35,36,37,38,39,40], and there is no consistent protocol for the therapy. Moore et al., 1998 [41] analyzed 73 patients with primary, advanced (FIGO III/IV) vulvar cancer and neoadjuvant chemoradiation with cisplatin/5-fluorouracil; 46.5% of the patients had complete remission and 53.3% had a gross residual tumor at the time of surgery, with positive resection margins in 13%. Additionally, two patients had unresectable residual tumors. Since in most cases of recurrent disease there are no possibility for further radiation and because of the limited amount of data available, neoadjuvant chemoradiation cannot be recommended unrestrictedly in cases of recurrent vulvar cancer. However, if feasible, neoadjuvant radio(chemo)therapy could even lead to higher surgical complication rates, as seen in cases with previous radiotherapy [42,43,44]. In cases of advanced disease with bladder or intestinal infiltration, radiation could lead to a higher risk for fistulas.

Therapeutic options for recurrent vulvar cancer not amenable to radiotherapy are even more limited without a standard for the chemotherapy regime. Chemotherapy has a proven benefit in combination with radiation (chemosensitization, with e.g., cisplatin, 5-fluorouracil, and mitomycin-C in lower doses) [45]. The benefit of chemotherapy as a solitary treatment is low. In earlier times, the substances used differed between cisplatin, paclitaxel, bleomycin, navelbin, and 5-fluorouracil, whether used alone or in combination. Due to the positive results in cervical cancer, platinum-based chemotherapy can be considered in the case of vulvar cancer. Here, carboplatin or cisplatin plus paclitaxel are preferred [46]. Recently, the combination of carboplatin/paclitaxel, as used in other gynecological cancers, has been applied to advanced vulvar cancer, which is less toxic but as effective as cisplatin/paclitaxel [45].

The overall response rate for solitary chemotherapy (neither in a neoadjuvant nor in an adjuvant setting) is 0–40%, with a PFS of 1–10 months and an OS of 19 months [18]. Due to a lack of randomized phase III studies, only limited data for systemic therapy in the treatment of vulvar cancer exists. For recurrent cancers, in particular, only non-randomized phase II studies with different chemotherapeutic regimens and with less than 50 patients are available [18]. For example, a phase II study of 31 patients with recurrent/metastatic vulvar cancer and chemotherapy with paclitaxel showed an overall response of 14% and a PFS of 2.6 months [47]. Another retrospective study of 16 patients with recurrent/metastatic vulvar cancer and chemotherapy with cisplatin plus vinorelbine showed an overall response rate of 40%, a median PFS of 10 months, and an OS of 19 months [48].

Overall, the data available shows that chemotherapy is less effective in vulvar cancer compared to other HPV-related tumors [34]. Due to this and because of the data described, no clear recommendation for solitary chemotherapy exists.

Nowadays, targeted therapies like epidermal growth factor receptor inhibitors or immune checkpoint inhibitors can be an alternative to classical chemotherapy. Woelber et al. [27] reported 15 patients with recurrent or adverse SCC of the vulvar receiving targeted therapy after previous platinum-based chemotherapy. Five patients received the EFGR-inhibitor erlotinib, nine patients received bevacizumab, and three patients received pembrolizumab. In the erlotinib group, stable disease and partial response were reached in 40% (two patients) each. One patient (20%) showed a progressive disease. In the bevacizumab group, two patients (22%) reached a complete response, three patients (33%) reached stable disease, and one patient (11%) reached a partial response. Three (33%) patients had a progressive disease. In the pembrolizumab group, one patient (33%) reached stable disease; the other two (66%) showed progression of the disease. Nine patients (60%) suffered from adverse events due to the treatment, mostly in grades 1 and 2.

Based on the KEYNOTE-158 study, which included 101 recurrent/metastatic vulvar carcinomas, the overall response rate for treatment with pembrolizumab was 10,9%, and the median duration of response was 20.4 months. The median OS was 6.2 months, and the median PFS was 2.1 months [49]. Based on the efficacy of pembrolizumab in PD-L1-positive cervical cancer, pembrolizumab is approved by the FDA and listed by NCCN-Compendia as a treatment option for metastatic/advanced, or recurrent cancer of the vulvar [50]. In the checkmate-358 study, 24 patients with recurrent/metastatic HPV-associated cervical (*n* = 19) and vulvar/vaginal (*n* = 5) cancer were treated with the checkpoint inhibitor nivolumab; the overall response rate was 20% and the disease control rate was 80%, regardless of the PD-L1 status [51]. Based on this data, the NCCN listed nivolumab for treatment of HPV-associated advanced and recurrent vulvar cancer [50].

Some patients with recurrent/metastatic vulvar cancer may benefit from immunotherapy, but further investigations are required. It can be reasonable to consider immune checkpoint inhibitors in cases of HPV-association or PD-L1-positive vulvar tumors, extrapolated from cervical cancer.

In summary, there is no distinct advice for treatment in cases of advanced recurrent vulvar carcinomas. Data for chemotherapy or radio(chemo)therapy are sparse. In advanced local recurrence, neither chemotherapy alone nor radio(chemo)therapy are usually a curative option, as the described data shows. In cases of gross tumor mass, the option for radiation is limited, especially in cases of tumor bleeding and subsequent anemia, because an adequate hemoglobin level is essential for effective radiotherapy. Additionally, there is a risk of fistula formation in cases of infiltration of the bladder or the intestine.

Because of the heterogeneity of the patient cohorts, a reasonable comparison between the described results due to PE and the results of the studies approaching radiation/radiochemotherapy is difficult. Randomized phase III studies comparing surgery and systemic therapy/radiotherapy are needed. However, the data presented shows that PE is a good option in cases of recurrent/advanced cancer, with comparable or even better data for OS.

Hence, in cases of advanced recurrent vulvar cancer with no option of radio(chemo)therapy or a high level of suffering and reduced quality of life, PE should be considered, as this method showed good results with acceptable morbidity [17]. The limitations of our study are the small number of patients and the missing analysis of data regarding quality of life. Thus, we are aware that our results are based on a small patient cohort and that larger analyses, preferably with data from multiple centers, are required to strengthen our results.

An advantage of our study is that the collective is described clearly with detailed information about the outcome of recurrent vulvar cancer only, while most other PE studies describe a heterogeneous collective of many different cancer types.

## 5. Conclusions

PE, whether for primary or recurrent vulvar cancer, offers a marked improvement in OS and is often the only chance for curation. In cases of advanced local recurrence and if radio(chemo)therapy is not an option, PE should be considered for vulvar cancer, as this method showed good results with acceptable morbidity.

## Figures and Tables

**Figure 1 cancers-16-00276-f001:**
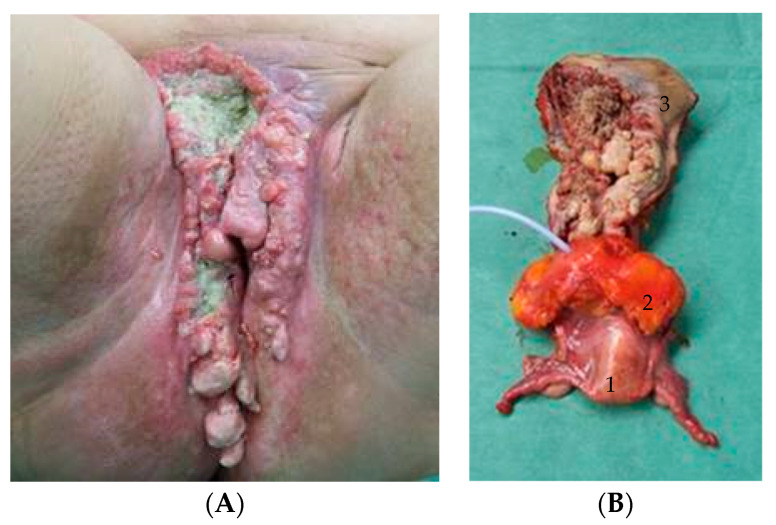
(**A**) View of an extended recurrent vulvar carcinoma with a uro-genital fistula formation. (**B**) Anterior PE preparation consists of uterus (1), bladder (2), and vulvar region (3).

**Figure 2 cancers-16-00276-f002:**
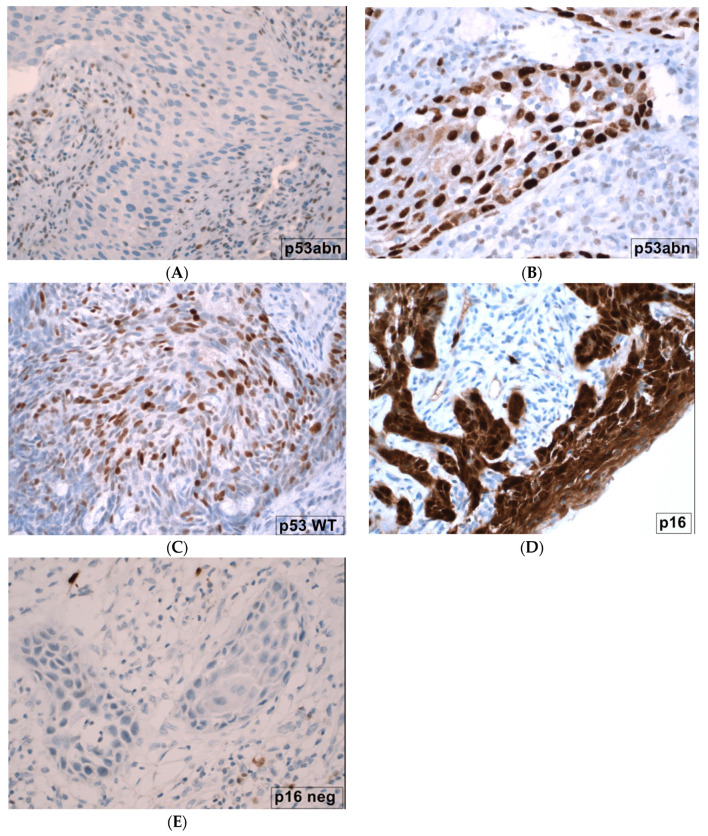
Microscopic images of p16- and p53-immunostainings in SCC vulvar cancer (magnification ×200). (**A**) p53 mutation; abnormal p53 staining, characterized by complete loss of expression of p53 in all tumor cells, indicative of a *TP53*-mutation. (**B**) p53 mutation; abnormal p53 staining, characterized by strong nuclear positivity (overexpression) of p53 in all tumor cells, indicative of *TP53*-mutation. (**C**) p53 wild-type: heterogenous basal/parabasal staining with variable intensities in the proliferating keratinocytes, wildtype-pattern. (**D**) p16 positive: HPV-associated, strong, block-like nuclear and cytoplasmatic positivity for p16. (**E**) p16 negative, HPV-independent.

**Figure 3 cancers-16-00276-f003:**
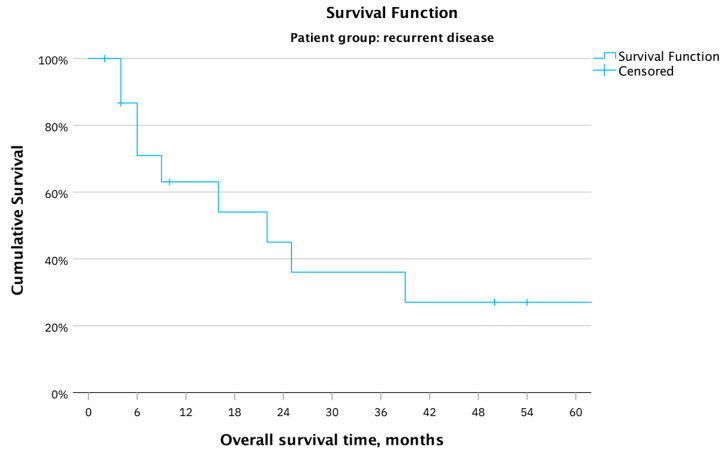
OS in months for the whole patient cohort (17 patients) with recurrent vulvar carcinoma after PE.

**Figure 4 cancers-16-00276-f004:**
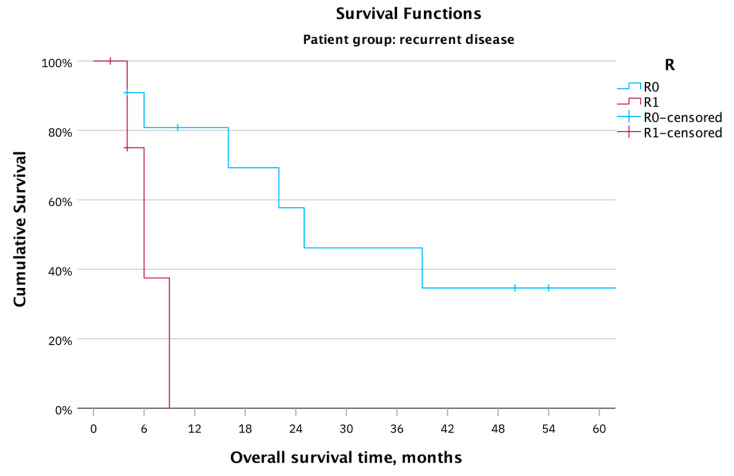
OS in months for the whole patient cohort (17 patients) with recurrent vulvar carcinoma after PE, based on R0/R1 tumor resection; complete cytoreduction revealed a statistically significant effect on the OS.

**Figure 5 cancers-16-00276-f005:**
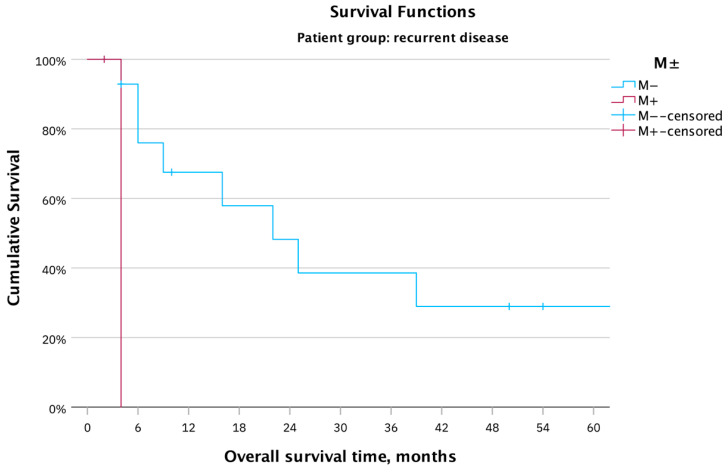
OS in months for the whole patient cohort (17 patients) with recurrent vulvar carcinoma after PE, based on distant metastases; the absence of distant metastases showed a statistically significant effect on the OS.

**Figure 6 cancers-16-00276-f006:**
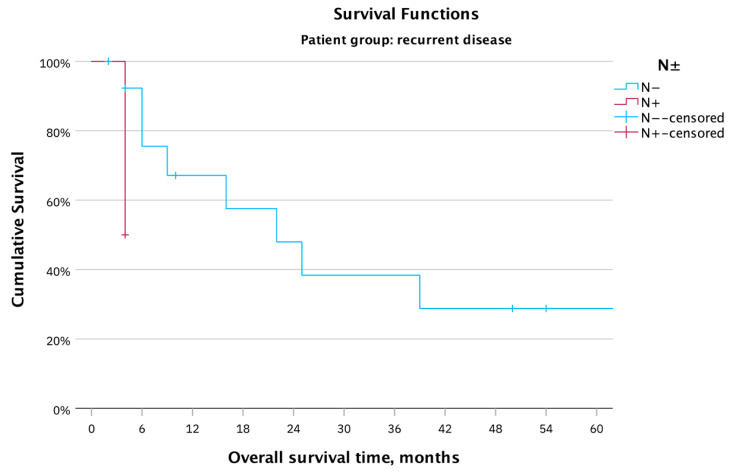
OS in months for the whole patient cohort (17 patients) with recurrent vulvar carcinoma after PE; based on lymphatic metastases, the absence of lymphatic metastases was a favorable variable for the OS.

**Table 1 cancers-16-00276-t001:** 17 cases of recurrent SSC with underwent PE; patients and tumor characteristics.

Patient	Age at Surgery (Years)	Type of PE	Number of Recurrence	T-Stadium	N-Status	M-Status	L-Status	V-Status	R-Status	Grading	Indication for Surgery	Neoadjuvant Treatment	Adjuvant Treatment	p16-Status	p53-Status
1	47	total	4th	rpT3	pN1	M0	L0	V0	R0	G2	curative	no	yes	negative	mutation
2	81	anterior	4th	rpT3	pNx	M0	L0	V0	R0	G2	curative	no	no	negative	mutation
3	70	anterior	1st	rpT3	cN0	M0	L0	V0	R0	G2	curative	no	no	negative	mutation
4	74	posterior	3rd	yrpT4	pNx	cM1 (PUL)	L0	V0	R1	G2	palliative	yes	unknown	negative	wildtype
5	70	total	1st	rpT3	pN0	cM0	L1	V1	R0	G3	curative	no	no	positive	wildtype
6	41	anterior	1st	rpT3	pN0	cM0	L0	V0	R1	G2	curative	no	yes	unknown	unknown
7	65	anterior	1st	rpT3	pN0	cM0	L0	V0	R0	G2	curative	no	no	negative	mutation
8	65	anterior	1st	rpT4	pN0	cM0	L1	V0	R1	G2	curative	no	no	negative	mutation
9	53	anterior	1st	rpT3	pN1	cM0	L1	V0	R1	G2	curative	no	no	negative	mutation
10	43	anterior	2nd	ypT2	pNx	cM0	L0	V0	R0	G2	curative	yes	yes	negative	mutation
11	77	anterior	1st	rpT2	pNx	cM0	L0	V0	R0	G2	curative	no	no	negative	mutation
12	70	total	2nd	rpT3	pN0	Mx	L0	V0	R0	G2	curative	no	no	positive	wildtype
13	72	anterior	1st	rpT3	pN0	pM1 (SKI)	L0	V0	R1	G2	palliative	no	unknown	negative	wildtype
14	63	posterior	1st	rpT2	pNx	cM0	L0	V0	R1	G2	curative	no	no	negative	mutation
15	78	posterior	3rd	rpT2	pNx	cM0	L0	V0	R0	G2	curative	no	no	negative	mutation
16	74	posterior	1st	rpT2	pNx	cM0	L0	V0	R0	G2	curative	no	no	positive	wildtype
17	61	anterior	1st	rpT3	pN0	cM0	L0	V0	R0	G3	curative	no	no	negative	mutation

**Table 2 cancers-16-00276-t002:** Results of the univariate Cox-analysis: independent variables with hazard ratio and confidence interval.

Independent Variable	Hazard Ratio [Confidence Interval]
Age at Surgery	0.98 [95% CI 0.931.04]
Blood Vessel Invasion	0.04 [95% CI non-applicable]
Complete Cytoreduction	6.3 [95% CI 1.02–39.03]
Complications (Clavien–Dindo classification ≤2/>2)	1.6 [95% CI 0.42–6.04]
Indication for Surgery	14.0 [95% CI 0.88–223.87]
Distant Metastases	14.0 [95% CI 0.88–223.87]
Lymphatic Metastases	0.19 [95% CI 0.41–103.92]
Lymphatic Vessel Invasion	1.18 [95% CI 0.13–10.66]
p16-status	0.46 [95% CI 0.06–3.67]
p53-status	1.04 [95% CI 0.21–5.05]
Tumor Grading	0.04 [95% CI 0.0–83.06]
T-Stadium (rT3 vs. rT2/rT4 vs. rT2)	1.09 [95% CI 0.27–4.48]/4.12 [95% CI 0.37–46.14]

**Table 3 cancers-16-00276-t003:** Postoperative complications, according to the Clavien–Dindo classification, in all 17 patients.

Patient No.	Complication According to the Clavien–Dindo Classification [12]
1	3b (postoperative defect coverage)
2	4a (postoperative dialysis)
3	2
4	1
5	1
6	1
7	1
8	3b (surgery anastomotic leakage ileum)
9	2
10	3b (postoperative defect coverage)
11	1
12	2
13	3b (wound debridement)
14	1
15	2
16	3b (ureteral stents)
17	1

**Table 4 cancers-16-00276-t004:** Four studies of PE in case of vulvar cancer exclusively, summary of the results.

Study	Study Type	Patient Group	Number of Patients	Median Age (Years)	5-Year OS	Factor Influencing OS
Valstad et al., 2023 [19]	single center; retrospective	primary locally advanced (53%), recurrent (47%) vulvar cancer	30	66	50%	None
Abdulrahman et al., 2022 [20]	single center; retrospective	primary locally advanced (74%), recurrent (26%) vulvar cancer	19	65	all: 66.7% primary disease: 69.3% recurrent disease: 60%	primary: lymphovascular invasion recurrent: none
Forner et al., 2011 [22]	single center; retrospective	primary locally advanced (33%), recurrent (64%) vulvar cancer	27	66	all: 62% primary disease: 62% recurrent disease: 59%	lymph node involvement, resection status (R0)
Hopkins et al., 1992 [21]	single center; retrospective	primary locally advanced (58%), recurrent (42%) vulvar cancer	19	50	60%	lymph node involvement

## Data Availability

Data are contained within this article.

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
