# Peer review of "Pelvic Exenteration for Recurrent Vulvar Cancer: A Retrospective Study"

_cancers, 2024, doi:10.3390/cancers16020276_

Round 1

Reviewer 1 Report

Comments and Suggestions for Authors

The title is well chosen and attractive, captivating reader's attention, being also relevant for the content presented. Introduction provides the necessary background of the subject and underlines the importance of this particular study. The study design is correct and the results presented are clinically relevant, despite the retrospective nature of the analysis and the relatively small number of patients included. Discussion is exhaustive, based on important references and also includes the clinical significance of the results of this study, raising awareness about recurrent vulvar cancer radical surgical management.

Minor editing  is necessary.

3.3 Complications - line 232 replace "mayor" to "major"; te same in lines 332 and 357

Comments on the Quality of English Language

 Minor editing of English language required

Author Response

Dear Reviewer 1,

Thank you very much for taking the time to review our manuscript. We appreciate your positive comments about our article. Following your recommendations, we edited your language revisions and examined the whole manuscript for English language.

Our team believes that pelvic exenteration is a relevant treatment option and offers a prognostic improvement for patients with recurrent vulvar cancer. We hope our study contributes to improve the highly limited treatment options for these patients.

Sincerely,

S. Claßen-von Spee

Reviewer 2 Report

Comments and Suggestions for Authors

Thank you for submitting this paper to cancers.
1- please review the English language, as there are many failures in the text (for Example: Line 37, Mayor in stead of major; Line 68, a few studies and in the table jes in stead of yes!)

2- Why did you exclude the other histological subtypes from this study? please explain it.

3-Can you please explain, how did you define the curative indication

4- Do not you have any data about the reconstructive surgery (vulva or vagina)  after the procedures?

Comments on the Quality of English Language

many spilling failures so that the language has to be reviewed professionally.

Author Response

Dear Reviewer 2,

Thank you very much for taking the time to review our manuscript. We are glad that you liked our article. Following your recommendations, we edited your language revisions and examined the whole manuscript for English language. Please find the responses to your comments below and the corresponding revisions highlighted in yellow in the revised manuscript.

2- Why did you exclude the other histological subtypes from this study? please explain it

R: As SCC is the most common type in vulvar cancer, we excluded other histological subtypes to keep the cohort homogenous.

3-Can you please explain, how did you define the curative indication.

R: If R0-resection seemed achievable the indication was defined as curative.

4- Do not you have any data about the reconstructive surgery (vulva or vagina) after the procedures? 

R: Seven patients received reconstructive surgery of the vulvar region.

We appreciate your comments about our article as they helped to further specify it. Our team believes that pelvic exenteration is a relevant treatment option and offers a prognostic improvement for patients with recurrent vulvar cancer. We hope our study contributes to improve the highly limited treatment options for these patients.

Sincerely,

S. Claßen-von Spee

Reviewer 3 Report

Comments and Suggestions for Authors

Dear authors,

this is an interesting paper on pelvic exenteration for recurrent vulvar cancer. However the small patient population limit the strength of the study. Some improvement are needed to consider it for publication.

Simple summary: line 15 please be more stringent, avoid “40-50%”, i would prefer “almost half”

Line 50-52 I would also state that stage at presentation and number of positive lymph nodes has not changed over a long time (as reference: 10.1038/s41598-021-85030-x)

While, about the Role of immunotherapy in vulvar cancer: 10.1016/j.gore.2022.100982

, to add some interesting data.

On the role of preinvasive vulvar lesions, I would also some lines on presence of VHSIL and dVIN, as the presence of lesions may impact on the overall survival. As the most recent ESGO guidelines, it should be stated the presence of VHSIL or dVIN in your case series (10.1097/LGT.0000000000000683)

Thank you for your precious work

Comments on the Quality of English Language

Minor editing

Author Response

Dear Reviewer 3,

Thank you very much for taking the time to review our manuscript. We are pleased about your positive comments about our article.

Please find the responses to your comments below and the corresponding revisions highlighted in yellow in the revised manuscript. Thank you very much for your helpful suggestions and the corresponding references. You pointed out a couple of very interesting aspects. In the final version we added the data about the involvement and number of positive lymph nodes and further data about the role of immunotherapy in vulvar cancer, as you recommended. We also added a paragraph about VHSIL and dVIN and the recommendations for treatment following the ESGO guidelines. Unfortunately, we do not have any data about precursor lesions in our patients since we treated them in case of recurrent invasive carcinoma. None of them showed precursor lesions at the time of surgery. Your point is highly interesting and should be considered in further investigations.

In addition, we examined the manuscript for English language.

We appreciate your comments as they helped to further specify this article. Our team believes that pelvic exenteration is a relevant treatment option and offers a prognostic improvement for patients with recurrent vulvar cancer. We hope our study contributes to improve the highly limited treatment options for these patients.

Sincerely,

S. Claßen-von Spee

Round 2

Reviewer 3 Report

Comments and Suggestions for Authors

Dear authors,

the manuscript has been improved according to previous round of review.

Now the paper deserves publication

Congratulations for your hard work